# Behavioral Intention in Domestic Heritage Tourism—An Extension of the Theory of Planned Behavior

Peter Onyonje Osiako [1,2,*] and Viktória Szente [3,*]

1 Doctoral School of Management and Organizational Sciences, Hungarian University of Agriculture and Life Sciences, Kaposvár Campus, Guba S. u. 40, 7400 Kaposvár, Hungary

2 Institute of Tourism and Hospitality Management, Dedan Kimathi University of Technology, Private Bag Dedan Kimathi, Nyeri 10143, Kenya

3 Institute of Agriculture and Food Economics, Hungarian University of Agricultural and Life Sciences, Kaposvár Campus, Guba S. u. 40., 7400 Kaposvár, Hungary

* Correspondence: peterosiako78@gmail.com (P.O.O.); szente.viktoria@uni-mate.hu (V.S.)

**Abstract:** Heritage tourist attractions are important in the diversification of tourism product offers for any competitive tourist destination. The current status of domestic heritage tourism in Kenya remains under-researched leaving many critical areas of interest requiring more research attention and redress. These needs also have to be examined in the Kenyan domestic heritage tourism context with a view to creating and satisfying higher demand for tourists. This research expands the widespread theory of planned behavior (TPB) by adding motivation and perceived safety and security as predictors of visit intention. In total, 802 respondents filled out the structured questionnaire, of which 693 questionnaires (86%) were found to be valid. The findings of this study validated the TPB with respect to heritage visitation in a domestic tourism context. It was further established that the expanded TPB model and its variables were applicable and more efficacious in directly predicting visit intention to historical heritage attractions. These results reveal the key determinants of the willingness of domestic tourists to visit historical heritage sites. Destination marketers and managers should endeavor to enhance the five psychographic aspects considered in the current study, in order to cultivate higher intentions towards visiting historical heritage sites among domestic tourists.

**Keywords:** heritage sites; Kenya coast; motivation; perceived safety; survey; structural model evaluation (SEM); willingness to visit historical sites

## 1. Introduction

Heritage tourism involves traveling to destinations of historical importance where historic events occurred, and places where interesting and significant cultures stand out [1]. It is perhaps the oldest form of tourism in the world and continues to dominate the tourism industry in many parts of the world [2]. To meet and satisfy ever-growing tourism demand, destinations need to develop new but sustainable products from the available resources and in critical consideration of market trends [3,4]. History-based tourism is now one of the most popular and globally widespread forms of special interest tourism. This is after becoming well established by the 19th century and increasing dramatically in the second half of the 20th century [5]. This form of tourism is overwhelmingly prominent in Europe (mainly Italy, Germany, and Spain) and Asia (mainly China), compared to Africa and the Arab States [6].

In recent times, the need to investigate the antecedents of tourist behavioral intentions and its relations with the preceding factors has attracted the attention of many researchers [7]. The current study used the extended Theory of Planned Behavior (TPB) to explore these factors.

The resulting extended model includes a considerable number of variables and is considered comprehensive enough to explicitly explain the behavioral intention of domestic

heritage tourists, particularly in the Kenyan heritage tourism context. Thus, the main objectives of the study were:

1. To examine factors that influence the intention to visit heritage sites by domestic tourists.
2. To validate the theory of planned behavior in the context of domestic heritage tourism.
3. To extend the theory of planned behavior and test the extended version in the context of domestic heritage tourism.

## 2. Theoretical Background

### 2.1. The Theory of Planned Behavior in Heritage Tourism

The TPB was developed as an extension of the Theory of Reasoned Action (TRA) [8]. Into this theory, Ajzen added a construct referred to as "perceived behavioral control" (PBC) as a determining factor for both behavioral intention and the behavior itself. While the earlier theory (TRA) comprised attitudes and subjective norms, TPB introduced and added the concept of perceived behavioral control (PBC), which was originally defined as an individual's perception of the ease or difficulty of performing a particular behavior [9]. PBC is thus deemed influential in determining whether the individual will engage in the behavior or not. The inclusion of this variable has been found to increase accuracy in predicting behavior that is not under volitional control [10].

Based on the TPB, intentions for (willingness to perform) a behavior are determined by three variables. The first variable is attitudes, which constitute an individual's overall evaluation of behavior. The second variable is subjective norms, which consist of a person's beliefs about whether significant others think he/she should engage in the behavior. The third variable measures the extent to which the individual perceives that the behavior is under their personal control and is labeled PBC. Ajzen [8] and Madden et al. [10] reported empirical evidence that PBC significantly improves predictions of both intentions and behavior. Hence, the evidence is broadly supportive of the TPB in helping to understand and predict behaviors, including travel and tourism behavior.

In the TPB, intention is the immediate precursor of behavior, and is assumed to be based on attitude toward the behavior, perceived social pressure or subjective norms, and perceived behavioral control. A significant number of empirical researchers in tourism have shown that the TPB efficaciously predicts tourist behavior in various tourism contexts [11–20]. However, the TPB does not take into account some important variables, such as the motivation that initiates tourists' behavior in visiting places, and other perceptual factors. Therefore, the current study added motivation and perceived safety and security factors to study their individual and collective roles in tourists' visit behavior.

The intention of tourists' consumption behavior is regarded as an important research topic in tourism [21]. Travel intention denotes an individual's commitment to travel or intent to travel. It can be viewed as an outcome of a psychological process that leads to transforming travel motivation into behavior, thus, a travel action. Jang et al. [22] noted that only limited empirical research has previously attempted to investigate the role of intention in the travel motivation–behavior relationship, leaving intention as one of the least researched areas of tourism. Iso-Ahola [23] associated leisure behavior with attitude, while Qu and Ping [24] assessed the link between the intention of Hong Kong residents to undertake cruise tours and their motivation. In both cases, a positive relationship was established. Shim et al. [25] also conducted a study and found that a more positive tourists' affective attitude corresponded to a stronger intention towards future travel. Separately, Hennessey et al. [26] attributed the intention to travel to two major elements of tourism marketing: responses to advertising and the respondent's use of official tourism websites.

The above studies suggest that travel intention may be influenced by multiple factors, including motivations, attitude, and promotion. Beldad and Hegner [27] have noted that new explorations continue to be focused on understanding how demographic characteristics, motivation, and cultural factors can also influence intention. The theory of planned behavior (TPB) introduced later by Ajzen [8] added the third predictor of behavioral inten-

tion to the TRA, calling it perceived behavioral control (PBC). This theoretical context has been applied in the understanding of travel intention as a kind of behavioral intention. It is, therefore, clear that in the psychology of human behavior, behavioral intention is widely acknowledged as the immediate antecedent to behavior, including travel behavior.

Attitude refers to "the degree to which a person has a favorable or unfavorable evaluation or appraisal of the behavior in question" [8] (p. 188). In consumer studies, it is described as the enduring, one-dimensional summary evaluation of a product or brand that is assumed to energize buying behavior [28]. Following the "principle of compatibility" [29], attitudes predict behavior. Ajzen and Fishbein [30] viewed attitude as a disposition to respond with some degree of favorableness or unfavorableness to a psychological object. According to them, attitudes are expected to predict and explain human behavior through behavioral intention, whereby positive attitudes should predispose approach tendencies, whereas negative attitudes should predispose avoidance tendencies. Travel attitude has been found to impact on tourists' intention to visit [31,32]. In applying this to consumer behavior studies, attitude and beliefs are also responsible for brand images formed in buyers' minds that affect their buying behavior [33].

Hence, it is hereby hypothesized that:

**Hypothesis 1. (H1a):** *Domestic tourists' attitude towards visiting historical heritage sites will positively influence visit intention towards historical heritage sites.*

Subjective norms are beliefs about the normative expectations of others that tend to exert perceived social pressure on an individual to have tendencies towards behaving, or actually behaving, in a certain manner [8]. This construct is widely considered in studies that apply the TPB, including travel and tourism research, for example [34–39]. However, some earlier tourism studies that followed the TPB model have found subjective norms not to have significant impact on leisure-related visit intention [36,37]. Hence, it is hereby hypothesized that:

**Hypothesis 2. (H2a):** *Domestic tourists' subjective norms relating to visiting historical heritage sites will positively influence visit intention for historical heritage sites.*

Perceived behavioral control denotes how people perceive the easiness or difficulty of performing a behavior of interest [8]. This factor is responsible for enabling or disabling the execution of behavioral goals. Many studies have supported the view that behavioral intention is produced from a combination of attitude toward the behavior, subjective norms, and perceived behavioral control (for example, [40,41]). According to TPB, the more favorable a person's attitude is towards a behavior and subjective norms, and the greater the perceived behavioral control, the stronger that person's intention will be to perform the behavior in question [40]. Therefore, the more people are able to have control over the opportunities and resources they have to perform a specific behavior, the more likely they will engage in such a behavior. Hence, it is hereby hypothesized that:

**Hypothesis 3. (H3a):** *Domestic tourists' perceptions of behavioral control as relates to visiting historical heritage sites will positively influence visit intention for historical heritage sites.*

### 2.2. Modifying TPB by Extension

The accuracy of the predictive ability of TPB for many different behaviors has been supported by many studies [42]. However, some other studies recommend adding more predictors to the theory, in order to increase its explanatory ability [43–46]. Therefore, this study integrated the additional factors of motivation and perceived safety and security into the proposed model, used to examine willingness to visit historical heritage sites. By integrating these additional variables into the TPB, the explanatory power of predicting

visit behavioral intention was expected to improve, without significantly affecting the three original TPB constructs, as explained below. Hence, Hypothesis H1b–H3b:

**Hypothesis 1. (H1b):** *In the new extended model, domestic tourists' attitude towards visiting historical heritage sites will positively influence visit intention for historical heritage sites.*

**Hypothesis 2. (H2b):** *In the new extended model, domestic tourists' normative belief as relates to visiting historical heritage sites will positively influence visit intention for historical heritage sites.*

**Hypothesis 3. (H3b):** *In the new extended model, domestic tourists' perceptions of behavioral control as relates to visiting historical heritage sites will positively influence visit intention for historical heritage sites.*

Motivation is one of the significant research topics covered in a number of studies in tourism research [47–49]. Most of these studies recognize the dynamism and heterogeneous nature of client motivation by considering tourist activities and individual personality relationships. Some authors, like [50], believe that, due to the dynamic concept of motivation, there is a possibility of identifying different tourist profiles based on these variables. Based on the social psychology point of view, the motives that compel a person to make a certain decision are closely connected to expectations which may result in great personal satisfaction [51]. Interestingly, literature on the choice behavior of tourists also indicates that motivation and "need" are interrelated [52]. They argue that tourists are attentive to stimuli that satisfy their desires and ignore stimuli that are not relevant to satisfaction of their desires and needs. Hsu and Huang [35] pointed to a paucity of research relating motivations and intentions to visiting tourist destinations and went ahead to discover that motivation had a positive influence on intention. Hence, it is hereby hypothesized that:

**Hypothesis 4. (H4):** *Domestic tourists' motivation to visit historical heritage sites will positively influence visit intention for historical heritage sites.*

The protection motivation theory (PMT) [53] proposed a modified version of expectancy-value theories, focusing on risk perception and change of intention. The theory postulates three crucial components of fear appeal: (i) the magnitude of the noxiousness of an environment; (ii) the probability of an event's occurrence; and (iii) the efficacy of a protective response. Protective motivation arises from these three components of fear appeal. Travel-related risks include, but are not limited to, cultural and language difficulties, natural disasters, terrorism, political instability, hygiene, diseases, crime and accidents, and environmental quality [54,55]. In relation to tourism, Sonmez and Graefe [56] established that an increase in cases of aviation accidents, crime, and terrorist activities represented danger and prompted careful selection of safe destinations, taking extra precautions while traveling to risky destinations, or canceling travel plans, among tourists. Destinations perceived as risky by potential tourists are avoided for those they consider safe. Buigut [57] has shown that terrorism has indeed significantly affected tourist arrivals and tourism earnings in Kenya. Hence, it is hereby hypothesized that:

**Hypothesis 5. (H5):** *Domestic tourists' perception of safety and security as relates to visiting historical heritage sites will positively influence visit intention for historical heritage sites.*

## 3. Materials and Methods

To a large extent this study adopted descriptive cross-sectional survey design, employing a quantitative approach. Self-administered paper and pen questionnaires were distributed by the researcher during the three months period of study, from December to March, which is high tourist season in Kenya (Season 2021/22).

We followed a two steps sampling method. First the sampling area was chosen, followed by the selection of target population. This research was conducted in three

counties of the Kenya coast tourism circuit (KCTC), comprising Mombasa County, Kilifi County, and Lamu county (see map of Kenya, Figure 1).

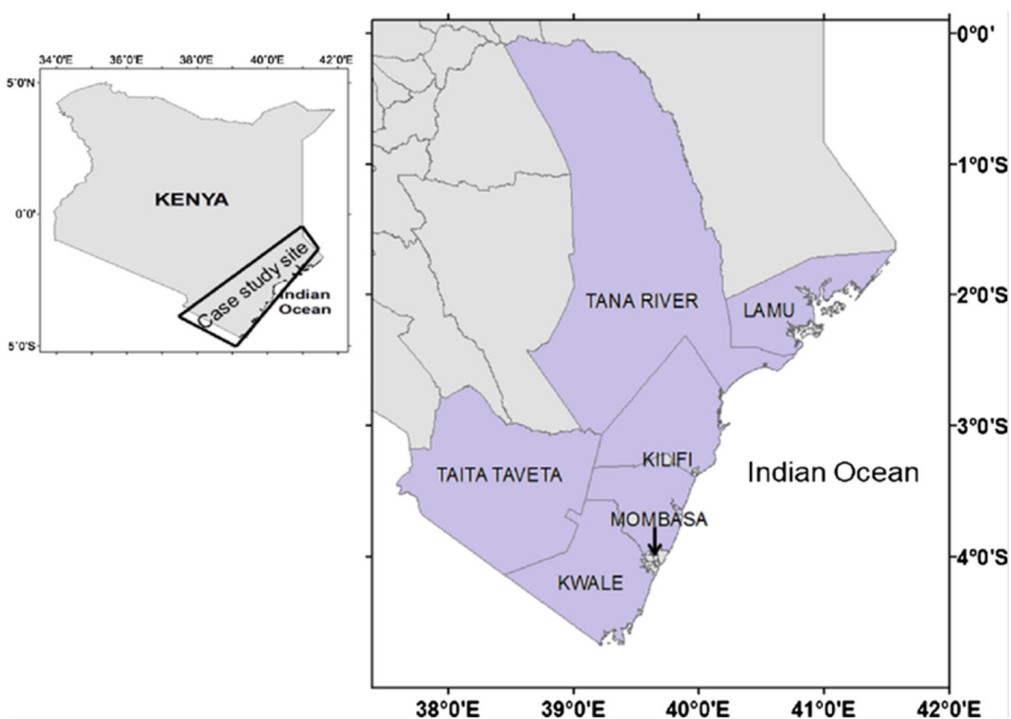

**Figure 1.** Map showing Kenya Coast Tourist Circuit. Source: https://www.google.com/search?q=kenya+map+coast+provinceandclient= (accessed on 8 November 2021).

KCTC was chosen for this study on the basis of its comparatively high number of tourist visitations in the country and the highest concentration of gazette historical heritage features. The National Museums of Kenya (NMK) has listed 142 gazette heritage features in this circuit under different categories (http://www.museums.or.ke/594-2/, (accessed on 8 November 2021)). However, it is important to note that, currently, not all these heritage features seem to be significant to tourism. This was confirmed by a short list of heritage sites in this tourist circuit that are usually visited by tourists, as captured by the Kenya National Bureau of Statistics (KNBS). The (Kenya) Economic Surveys of 2019 from KNBS figures revealed that the coastal region accounted for 43.1% (3,716,900 out of 8,617,900) of all the bed-night stays in the country in the year 2018 [58]. This makes this region the busiest tourist circuit in the country. Of all the tourists who visited museums, snake parks and monuments in the country during the same period (2018–2019), 35.2% of them visited sites in the KCTC.

The criteria for sample selection for this study involved those historical heritage sites that were both managed by the NMK and significant to tourism in terms of visitor numbers. According to the KNBS [58], there were 11 sites managed by NMK and significant to tourism in the KCTC.

The only target population for this study are domestic tourists who visited the heritage attractions. Questionnaires were administered systematically such that every third visitor who entered the heritage attraction was invited to take part in the survey. This was repeated for the three months on the three days of the week when domestic tourists frequented these sites. The respondents thoroughly read through the instructions, which gave a clear description of the survey and its purpose. A total of 802 questionnaires were received from respondents, indicating a response rate of 91%. After excluding the non-usable questionnaires, the final data set used for the analysis comprised of 693 (86%) respondents. Only fully completed questionnaires were analyzed, accounting for the lesser number that was ultimately analyzed.

Based on the constructs in the TPB and review of the relevant literature, the final structured questionnaire included 28 items which were Likert scale-type questions, 3 ratio scale type, and 3 open-ended question types. The rest of the questions were of a multiple-choice type and sought answers regarding the respondents' socio-demographic characteristics. The validity and reliability of these 31 psychometric items were confirmed through pilot testing and factor analysis.

The survey development process was as follows. There were nine parts to the survey: the first seven parts contained items designed to identify perceptual constructs (attitude, subjective norms, perceived behavioral control, perceived safety and security, and motivation), intention to visit historical heritage sites, and visit behavior measures. The eighth part asked about the general travel behavior of the respondents, and the last part contained questions seeking demographic information. Measurement items were developed based on an extensive literature review and previous studies that applied to the TPB. For the attitude, subjective norm, perceived behavioral control, and behavioral intention items, the TPB model was employed, mainly based on the suggestion of Fishbein and Ajzen [59]. As for motivation and perceived safety and security, the respective items were developed following previous conceptualizations and studies in the context of leisure tourism and modified as relating to the area under study. All these items were measured using a seven-point Likert scale while the visit behavior construct was measured on a seven-point ratio scale. In total, the final survey included 31 items.

The collected data was analyzed using the IBM SPSS Statistics for Windows version 23.0 (IBM Corp., Armonk, NY, USA) software. Thereafter, hypothesis testing using SPSS Amos for Windows version 26.0 was carried out to establish the association between the five hypothesized predictors and the intention to visit historic heritage sites. Both confirmatory factor analysis (CFA) and structural equation modelling (SEM) were used for hypothesis testing.

*Description of the Sample*

Of the total respondents, 54.5% were male and 44.9% were female. Those in the category of "other" represented 1.6%. 36.4% of respondents were aged 26–35 years. The majority were employed on a full-time basis (29.7%), and the majority earned an income of, at most, 10,000 Kenyan shillings (USD 100) per month (32.2%). The biggest proportion of the sample (56.1%) were middle-level college or bachelor's degree holders. A review of the data revealed that a majority (30%) of the respondents traveled to heritage sites quite irregularly and preferred visiting the coastal region of Kenya (84%). Heritage, history, and culture were their most preferred categories of attractions (48%), and most of them originated from the coastal region (43%) (Table 1).

**Table 1.** Respondents' socio-demographic profiles.

| Socio-Demographic Variable | | Frequency | Percent |
|---|---|---|---|
| Gender (N = 693) | Male | 378 | 54.5 |
| | Female | 304 | 43.9 |
| | Other | 11 | 1.6 |
| Age in years (N = 693) | 18–25 | 245 | 35.4 |
| | 26–35 | 252 | 36.4 |
| | 36–45 | 105 | 15.2 |
| | 46–55 | 66 | 9.5 |
| | 56–65 | 19 | 2.7 |
| | Over 65 | 6 | 0.9 |

| Socio-Demographic Variable | | Frequency | Percent |
|---|---|---|---|
| Your income in KES (N = 693) | 10,000 and below | 223 | 32.2 |
| | 10,001–25,000 | 162 | 23.4 |
| | 25,001–50,000 | 143 | 20.6 |
| | 50,001–100,000 | 98 | 14.1 |
| | 100,001–200,000 | 36 | 5.2 |
| | over 200,000 | 31 | 4.5 |
| Your marital status (N = 693) | Not in Marriage | 348 | 50.2 |
| | Married without Children | 119 | 17.2 |
| | Married with Child/ren | 226 | 32.6 |
| Highest educational level attained (N = 693) | No formal Education | 22 | 3.2 |
| | Primary | 29 | 4.2 |
| | Secondary | 134 | 19.3 |
| | College/Bachelor's degree | 389 | 56.1 |
| | Post Graduate Degree | 119 | 17.2 |
| Region of origin in Kenya (N = 651) | Coast | 280 | 43 |
| | Eastern | 31 | 4.7 |
| | North Rift Valley | 31 | 4.7 |
| | Nairobi | 137 | 21 |
| | Central | 88 | 13.5 |
| | South Rift Valley | 13 | 2 |
| | Western | 25 | 3.8 |
| | Nyanza | 40 | 6 |
| | North Eastern | 6 | 0.9 |
| Employment status (N = 693) | Self Employed | 154 | 22.2 |
| | Employed Full Time | 206 | 29.7 |
| | Employed Part Time | 57 | 8.2 |
| | Seeking Opportunities | 121 | 17.5 |
| | Retired | 21 | 3.0 |
| | Student | 109 | 15.7 |
| | Home Maker | 14 | 2.0 |
| | Unable to Work | 4 | 0.6 |
| | Other | 7 | 1.0 |

Source: Researchers' data analysis.

## 4. Results

This section presents the processes and the results of the CFA and SEM analyses. As a multivariate technique, SEM is used for modeling tests that include several independent and dependent constructs, as well as mediators and moderators. Through SEM, researchers are able to concurrently assess a series of dependent relationships [60].

### 4.1. Exploratory Factor Analysis

To assess the dimensionality of the 31 item statements in the questionnaire relating to the variables under study, Exploratory Factor Analysis (EFA) was conducted. Then the factor loading values that indicated the correlation between items and factors were identified. These determined whether the group of observed variables could be presented by the factor or not. An Eigenvalue of greater than 1.0 was determined and items with factor loadings greater than 0.7 were taken for each factor grouping (Table 2). Cronbach's alpha ($\alpha$) was applied to test reliability of factor groupings. The factors with Cronbach $\alpha$ greater than 0.6 were taken for analysis. Cronbach's alpha coefficient was calculated to evaluate the internal consistency.

First, the suitability of the data was assessed through an exploratory factor analysis of the 31 statements related to the variables under study. Factor analysis with a Principal Component Approach and Promax rotation with Kaiser Normalization were conducted. KMO Bartlett's test was carried out to verify the normality and significance of the conducted analyses and it was found to be highly significant (approximate $\chi^2$ = 7083.388, df = 300, $p < 0.05$). Bartlett's Test of Sphericity ($\chi^2$ = 7333.790) and the Kaiser–Meyer–Olkin (KMO)

overall measure of sampling adequacy (0.886) indicated that the data were suitable for using factor analysis [61].

**Table 2.** Exploratory factor analysis, reliability and validity tests for variables.

| Items | Mean | Factor Loading | Eigenvalue | Cronbach Alpha | AVE (>0.5) | CR (>0.7) | *p*-Value |
|---|---|---|---|---|---|---|---|
| **Attitude** | 5.89 | | 7.455 | 0.845 | 0.561 | 0.885 | |
| Bad or good idea | | 0.704 | | | | | 0.000 |
| Desirability | | 0.741 | | | | | 0.000 |
| Enjoyability | | 0.758 | | | | | 0.000 |
| Pleasantness | | 0.801 | | | | | 0.000 |
| Rewarding or not rewarding | | 0.758 | | | | | 0.000 |
| Usefulness | | 0.733 | | | | | 0.000 |
| **Motivation** | 5.72 | | 1.226 | 0.678 | 0.598 | 0.817 | |
| Education/personal knowledge | | Dropped | | | | | |
| Recreation and enjoyment purposes | | 0.788 | | | | | 0.000 |
| Cultural purposes | | Dropped | | | | | |
| Socialization purposes | | 0.707 | | | | | 0.000 |
| Adventure purposes | | 0.821 | | | | | 0.000 |
| Boosting my self-esteem | | Dropped | | | | | |
| Subjective Norm | 5.12 | | 1.405 | 0.862 | 0.768 | 0.908 | |
| Most people who are important to me think that it is proper for me to visit HHS at the Kenyan Coast | | 0.874 | | | | | 0.000 |
| Most people who are important to me would want me to visit HHS at the Kenyan Coast | | 0.890 | | | | | 0.000 |
| People whose opinions I value would prefer that I visit HHS at the Kenyan Coast | | 0.865 | | | | | 0.000 |
| **Perceived Safety and Security** | 5.68 | | 1.543 | 0.853 | 0.757 | 0.903 | |
| HHS are safe and secure places to visit | | 0.893 | | | | | 0.000 |
| I feel safe and secure at HHS | | 0.901 | | | | | 0.000 |
| There are no risks at HHS | | 0.814 | | | | | 0.000 |
| **Perceived Behavioral Control** | 5.28 | | 1.660 | 0.779 | 0.577 | 0.845 | |
| Visiting HHS is my decision | | Dropped | | | | | |
| Whenever I want, I visit HHS | | Dropped | | | | | |
| I have financial resources to visit HHS | | 0.742 | | | | | 0.000 |
| I can easily spare time to visit HHS | | 0.746 | | | | | 0.000 |
| I have sufficient information to decide on visiting HHS | | 0.791 | | | | | 0.000 |
| I can conveniently convenient HHS at the Kenyan Coast | | 0.759 | | | | | 0.000 |
| **Intention** | 6.00 | | 2.226 | 0.791 | 0.590 | 0.858 | |
| I have the intention to visit HHS | | 0.741 | | | | | 0.000 |
| I will make an effort to visit an HHS in the next year | | 0.811 | | | | | 0.000 |
| In future, I will re-visit an HHS | | 0.778 | | | | | 0.000 |
| I am willing to recommend HHSs | | 0.773 | | | | | 0.000 |
| **Visit behavior** | 4.11 | | 1.067 | 0.638 | 0.500 | 0.829 | |
| Previous 1-year visits to HHS attractions in the Kenyan Coast region | | 0.860 | | | | | 0.000 |
| Likelihood to frequent HHS | | Dropped | | | | | |
| Future 1-year visits to HHS | | 0.823 | | | | | 0.000 |

Note: AVE *—Average variance explained, CR **—Composite Reliability.

Ultimately, seven factors were identified which accounted for 66.330% of total variance, that is, 29.822%, 8.902%, 6.642%, 6.173%, 5.621%, 4.904%, and 4.266% for attitude, intention, PBC, safety, subjective norm, motivation, and visit behavior, respectively. The respective Eigenvalues were 7.455, 2.226, 1.660, 1.43, 1.486, 1.226, 1.067. Three items failed to adequately load on the Motivation variable and two failed to load on the PBC variable and were thus dropped. The Cronbach's alpha coefficient results obtained were 0.845, 0.678, 0.862, 0.853, 0.779, and 0.791, and 0.638, respectively, for the seven variables. These coefficients, together with AVE of 0.5 and above and CR of above 0.7, indicated that the items had internal consistency, were reliable, and valid [62].

### 4.2. Results of Correlation Analysis

Before starting correlation analysis, descriptive analysis was used. The values for Skewness and Kurtosis of between +1 and +2, respectively, indicated that the data related to the constructs were normally distributed and thus allowed for parametric statistics.

Pearson's correlation analysis revealed that all the seven constructs significantly correlated with each other (Table 3). All the 15 correlations were at moderate level and significant at $p < 0.01$. The highest correlation was between perceived behavioral control and subjective norm ($r(691) = 0.457$, $p < 0.01$), followed by that between perceived behavioral control and perceived safety and security ($r(691) = 0.448$, $p < 0.01$).

**Table 3.** Inter-construct correlation.

|       | Mean   | SD      | ATT      | INT      | PBC      | PSS      | SNM      | MOT |
|-------|--------|---------|----------|----------|----------|----------|----------|-----|
| ATT   | 5.8935 | 0.91519 | -        |          |          |          |          |     |
| INT   | 6.0025 | 0.93120 | 0.381 ** | -        |          |          |          |     |
| PBC   | 5.2781 | 1.21398 | 0.314 ** | 0.425 ** | -        |          |          |     |
| PSS   | 5.6821 | 1.25047 | 0.419 ** | 0.365 ** | 0.448 ** | -        |          |     |
| SNM   | 5.1236 | 1.41033 | 0.370 ** | 0.389 ** | 0.457 ** | 0.399 ** | -        |     |
| MOT   | 5.7225 | 1.08506 | 0.408 ** | 0.352 ** | 0.333 ** | 0.297 ** | 0.403 ** | -   |

** Correlation is significant at the 0.01 level (2-tailed). ATT—Attitude, INT—Intention, PBC—Perceived behavioral control, PSS = Perceived Safety and Security, SNM—Subjective norm, MOT—Motivation.

### 4.3. Validity and Reliability of Constructs

Apart from pilot testing and factor analysis, validity and reliability of constructs were determined. Convergent validity of a construct refers to how closely a used scale is related to other variables and other measures of the same construct. It is achieved when the calculated composite reliability (CR) is greater than 0.70, and when AVE is greater than 0.5. Discriminant/divergent validity of a construct shows that the construct is not correlated with dissimilar, unrelated others. It is achieved in three measures: when the square root of the AVE is greater than the correlation between the constructs [63], when AVE is greater than MSV, and finally when AVE is greater than ASV.

In Table 4, it can be seen that the AVE values are 0.5 and above, and they are above the correlation coefficients for each of the constructs. Equally, the square-roots of AVE are higher than inter-construct correlations, and also higher than MSN and ASV. Cronbach $\alpha$ is higher than 0.6, while composite reliability is higher than 0.7. Hence, the constructs and measures are both reliable and valid [60].

Following a multivariate assessment for data normality, a Confirmatory Factor Analysis (CFA) was employed to generate the measurement model. The CFA results of the data set indicated that the model fitted the data well ($\chi^2 = 690.369$, df = 254, $\chi^2/\text{df} = 2.718$, RMSEA = 0.050, CFI = 0.937 and TLI = 0.925). These results provided evidence for the uni-dimensionality of each scale. Measurement items were all significantly loaded to their associated factors ($p < 0.001$). In addition to the CFA, the AVE and CR for all measures were assessed for uni-dimensionality, reliability, and construct validity, as Hair et al. (2010) recommended. The values for composite reliability ranged from 0.817 to 0.908, exceeding the recommended threshold of 0.7 suggested by [60], indicating a high level of conver-

gent validity [60]. Finally, AVE values ranged from 0.561 to 0.768, thus exceeding the recommended value of 0.50 [60]. This confirmed convergent validity. Moreover, the AVE value for each variable was superior to the squared correlation between variables (Table 4), demonstrating that discriminant validity was attained [60].

**Table 4.** Reliability, convergent validity, discriminant validity and correlations.

| | CV | DV | | Reliability | | | | | | | | |
|---|---|---|---|---|---|---|---|---|---|---|---|---|
| | AVE | MSV | ASV | CR | $\alpha$ | ATT | INT | PBC | PSS | SNM | MOT | VBH |
| ATT | 0.561 | 0.166 | 0.125 | 0.885 | 0.845 | **(0.749)** | | | | | | |
| INT | 0.590 | 0.181 | 0.127 | 0.858 | 0.791 | 0.381 ** | **(0.768)** | | | | | |
| PBC | 0.577 | 0.209 | 0.138 | 0.845 | 0.779 | 0.314 ** | 0.425 ** | **(0.759)** | | | | |
| PSS | 0.757 | 0.201 | 0.130 | 0.903 | 0.853 | 0.419 ** | 0.365 ** | 0.448 ** | **(0.870)** | | | |
| SNM | 0.768 | 0.209 | 0.153 | 0.908 | 0.862 | 0.370 ** | 0.389 ** | 0.457 ** | 0.399 ** | **(0.876)** | | |
| MOT | 0.598 | 0.166 | 0.111 | 0.817 | 0.678 | 0.408 ** | 0.352 ** | 0.333 ** | 0.297 ** | 0.403 ** | **(0.768)** | |

** Correlation is significant at the 0.01 level (2-tailed). Note: Bold values in brackets and diagonal represent square root estimates of AVE. $\alpha$—Cronbach alpha, ASV—Average shared variance, AVE—Average variance extracted, CV—Convergent validity, CR = Composite reliability, MSV—Maximum shared variance, ATT—Attitude, INT—Intention, PBC—Perceived behavioral control, PSS = Perceived Safety and Security, SNM—Subjective norm, MOT—Motivation, VBH—Visit behavior.

*4.4. Structural Model Evaluation and Hypothesis Testing*

To test the measurement models and the research hypotheses, SEM was conducted. Generally, the fit indices indicated an adequate fit: for the conventional TPB ($\chi^2/df$ = 2.308, RMSEA = 0.043) and for the extended TPB ($\chi^2/df$ = 3.636, RMSEA = 0.066). Both these models included a satisfactory level of explanatory power for intentions to visit historic heritage sites. Some 47% and 49% respectively, of the total variance in domestic tourists' intention was accounted for by the three, and then by the five constructs, respectively, on the two models. After this, the hypotheses were tested.

The detailed results are reported in Table 5. As anticipated, in the first model, the results showed that attitude had a positive and significant influence on intentions to visit historical heritage attractions ($\beta$ = 0.27; $p < 0.001$), and subjective norms exerted a significant positive influence on these intentions ($\beta$ = 0.16; $p < 0.001$). Similarly, PBC had a positive and significant influence on intentions ($\beta$ = 0.41; $p < 0.001$). This result supported Hypotheses H1a–H3 (Figure 2). The model explained 47% of intentions to visit historic heritage sites.

**Table 5.** Summary of results of hypothesis testing.

| Model | Hypothesis | Relationship | Std $\beta$ | Decision |
|---|---|---|---|---|
| TPB Model ($R^2$ = 0.47) | H1a | Attitude $\rightarrow$ Intention | 0.27 *** | Supported |
| | H2a | Subjective norm $\rightarrow$ Intention | 0.16 ** | Supported |
| | H3a | PBC $\rightarrow$ Intention | 0.41 *** | Supported |
| Extended TPB Model ($R^2$ = 0.49) | H1b | Attitude $\rightarrow$ Intention | 0.16 *** | Supported |
| | H2b | Subjective norm $\rightarrow$ Intention | 0.10 ** | Supported |
| | H3b | PBC $\rightarrow$ Intention | 0.35 *** | Supported |
| | H4 | Motivation $\rightarrow$ Intention | 0.17 ** | Supported |
| | H5 | Perceived SS $\rightarrow$ Intention | 0.13 ** | Supported |

* $p > 0.05$, ** $p < 0.01$, *** $p < 0.001$.

In the second model representing the extended TPB, the results showed that all the five relationships between the predictors and visit intentions were positive and significant. Attitude had a positive and significant influence on intentions to visit historical heritage attractions ($\beta$ = 0.16; $p < 0.001$), and the same for subjective norms ($\beta$ = 0.10; $p < 0.01$), PBC ($\beta$ = 0.35; $p < 0.001$), motivation ($\beta$ = 0.17; $p < 0.01$), and finally perceived safety and security ($\beta$ = 0.13; $p < 0.01$). Therefore, Hypotheses H1b–H3b, H4 and H5 were supported (Figure 3). The new extended model explained 49% of intentions to visit historic heritage sites.

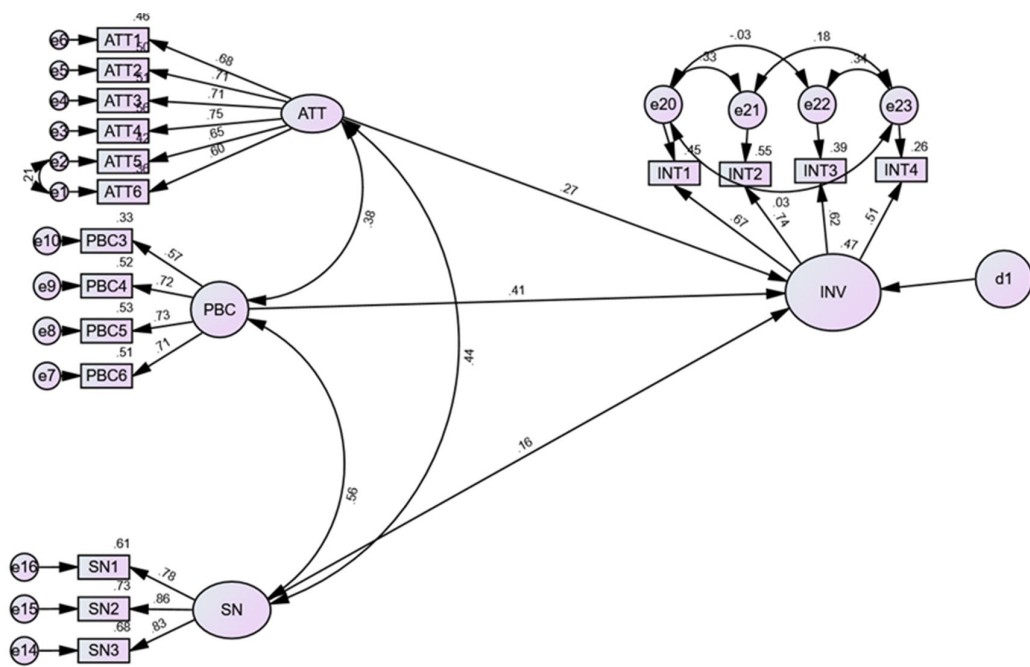

**Figure 2.** Determining visit intention using the traditional TPB model.

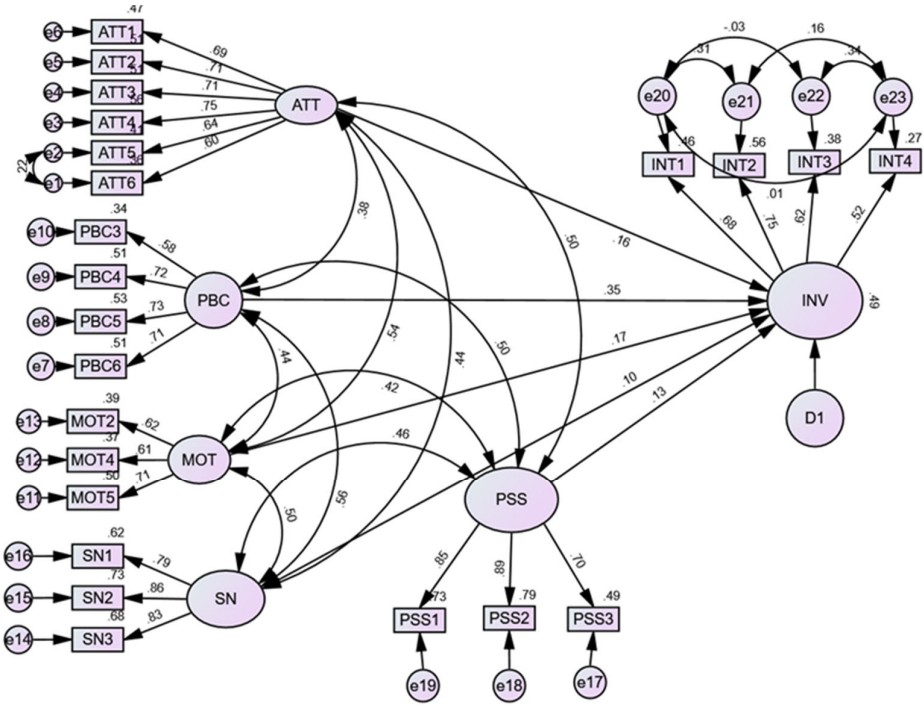

**Figure 3.** Determining visit intention using the extended TPB model.

Finally, *t*-tests indicated that there were no significant differences across age, education, marital status, employment status, and income level with respect to the effect of demographics on study variables. The analysis revealed that demographic factors had no bearing on the antecedents of the intentions to visit historic heritage attractions. The summarized results of hypothesis testing are presented in Table 5.

## 5. Discussion

### 5.1. Behavioral Intention and Its Antecedents

Intention is widely acknowledged as the immediate precursor of behavior [27,64]. These authors defined intention as an indication of an individual's readiness to perform a given behavior. It was therefore prudent, through the present study, to assess visit behavior to historical heritage attraction within the TPB framework and compare it to the expanded version. The study established that the associations in the TPB could also be applied in the heritage tourism context. It follows that an increase in prospective visitors' intention to visit historical heritage sites could directly enhance their actual behavior in visiting these heritage attractions [65]. The implication, therefore, is that the number of domestic tourists visiting heritage sites could be increased by cultivating higher visit intentions within the prospective domestic tourists.

Following the TPB model, increased visit intention for historical heritage attractions is anchored in three factors: a positive attitude towards historical heritage sites, a positive normative belief with regard to historical heritage sites and enhanced perceived behavioral control with regard to visiting historic heritage attractions. The TPB places a high premium on these three variables as predictors of intention. A positive correlation between these variables and visit intention with respect to historical heritage sites implies that a higher score in any of these variables translates to increased visit intention. Each of these three predictor variables in the model gives a unique but inter-dependent contribution to the outcome variable (visit intention), which ultimately determines visit behavior.

### 5.2. Theoretical Implications

In view of the need for further research in domestic tourism and heritage tourism, as pointed out by Light [5] and Jørgensen [66], the current research makes critical contributions to this area of tourism research. First, this study provides key implications with regard to volitional factors that impact domestic heritage site visit intentions. It reveals that attitudes have a remarkable impact on tourists' intentions. Attitude in this context constitutes tourists' favorable or unfavorable assessment of domestic heritage tours [67]. The relationship between tourist attitudes and their purchase intention is widely reported in previous studies [68–70]. The crucial role of attitude is equally important in marketing tourism, to the extent that attitude and beliefs are said to be responsible for brand images formed in buyers' minds that affect their buying behavior [33]. Considering that attitude is built through evaluation of the target behavior, tourism promotional strategies should be geared towards communicating the appeal of heritage attractions and the thrill that tourists can find therein. By so doing, marketers will leverage pragmatism regarding the attitude factor in order to elevate the stature of heritage tourism through aggressive marketing to influence visit intention.

Second, perceived behavioral control emerged as a strong predictor of the intention to visit historical heritage sites. This result relates to prior research regarding the role of PBC in the TPB model (e.g., [71,72]). Recent times have witnessed a growing research interest in the influence of PBC on tourists' consumption behavior and it has been established that, to varying degrees, PBC prompts the translation of tourists' visit intentions into actual visit behavior [71,73]. This implies that domestic tourists who felt confident that they had control over visiting historical heritage sites were more likely to do so, rather than those who lacked confidence and opportunities to do so. Ajzen and Kruglanski [74] and De Leeuw et al. [75] strongly associated perceived behavioral control with the degree to which one believes that conducting a certain behavior is under one's volitional control. Therefore, it is predicted that the perceived cumbersomeness (or ease) of visiting historical heritage sites might significantly influence the possibility of tourists engaging in this behavior. These findings suggest that domestic tourists' intentions towards visiting historical heritage sites could be activated by the availability of sufficient resources and facilitation. This further underscores the significance of making conditions that facilitate domestic heritage tourism through assisting in overcoming any perceived hindrances.

Similarly, subjective norms were found to influence tourists' heritage site patronage intention, as reported in some previous studies that investigated tourist behavior [34–38,76]. This noteworthy finding implies that domestic tourists fall under the significant influence of their referents' expectations and pressure to visit historical heritage sites. Contrary to this, Sparks [36] and Shen et al. [37] found that subjective norms did not have a significant impact on leisure-related visit intention. A possible explanation for their divergent findings was given as a failure to use suitable words when measuring subjective norms in the tourism context, especially when adapting them directly from Ajzen's [8] proposed statements. Shen et al. [37] has indicated that subjective norms have no significant correlations with the other two constructs in the TPB model, the possible reason why Shen [77] completely omitted this variable in his study. However, these two discrepancies did not affect the current studies.

In extending the TPB model for this particular study, motivation, which is "defined as the driving force that determines tourist behavior", proved to be a significant predictor of visit intention to historical heritage sites. Its addition improved the predictive power of the regression model. Related to this finding, a significant number of studies have, over the last fifty years, directly or indirectly studied motivations for tourism travel, starting with Plog [78] and continuing with others (e.g., [48,52,54,79]). However, only a few of these studies have attempted to address the relationship between motivational factors and behavioral intentions [20,35]. This study, therefore, extended the TPB model as proposed by Pearce and Packer [80], thereby elevating the behavioral intention to visit heritage sites. It then becomes evident that promotion of heritage tourism could be achieved by employing strategies aimed at increasing travel motivation for prospective domestic tourists.

It has been argued that perception of risk may lead to a change of intention [53]. The five hypotheses for this study sought to find out the influence of positive safety and security perception on behavioral intention for visiting historical heritage sites. A significant positive association was established in this hypothesis test, implying that higher visit intentions to heritage attractions could be exhibited by domestic tourists if they perceived the places to be safer and secure. On the contrary, associating these attractions with risk and insecurity could only serve to discourage visits to these attractions [32]. This finding agrees with other studies that have shown that safety and security are paramount for tourist destinations if larger numbers of visitors are to be expected [55]. With a particular reference to Kenya as a destination, Buigut [57] observed that terrorism had significantly affected tourist arrivals and earnings in Kenya. Therefore, in general, accidents, crime, diseases and terrorist activities represent danger that prompts careful selection of safe destinations by tourists, or even their canceling of travel plans. Tourists avoid destinations perceived as risky for those they consider safe.

Hence, with respect to the second objective, this study effectively tested and validated the efficacy of the traditional TPB. These findings were consistent with previous research in several other tourism contexts (e.g., [32,35,76]) which applied to the TPB. However, the said studies only tested the impacts of the three TPB constructs ('attitude', 'subjective norm' and 'perceived behavioral control') on 'tourists' behavioral intention'), or added only motivation or other variables separately, without including these additional variables at the same time.

Therefore, to achieve the third objective, the present study tested the extended TPB in the context of domestic heritage tourism by adding motivation and perceived safety and security simultaneously. As alluded to by Hagger et al. [42], the accuracy of the predictive ability of TPB for many different behaviors has been supported by many studies. However, some other studies recommend adding more predictors to the TPB in order to increase its explanatory ability [43,44,46]. This study integrated the additional factors of motivation and perceived safety and security in the proposed model used to examine the willingness to visit historical heritage sites. By integrating these additions into the TPB, the explanatory power of predicting visit behavioral intention was enhanced. Notably, the coefficient of determination rose from $R^2 = 0.47$ to $R^2 = 0.49$. The resulting model proves not only to

be more comprehensive, but also more efficacious in predicting domestic tourists' visits to historic heritage attractions. The implication is that the model for determining visit intention to heritage tourist attractions can be improved by including motivation, safety, and security in its perception. The outcome of the current study indicated that, the more predictors, the stronger and more stable the model. The advantage of multiple predictors is that, if one of the predictors falls short, it is easier to manipulate the other available predictors of visit intention to achieve the desired effect.

One other significant contribution of this study to academics is its unique context. The paper reports on the applicability of the extended TPB model based on a non-Western society and in a developing country, unlike most of the travel behavior models. The fact that Kenya has a different social, economic, political and cultural background means that its domestic tourists are a sample with distinctive characteristics and worthy of comparison with Western societies and others.

### 5.3. Practical Implications

The outcome of this study has several key implications for practice in a domestic heritage tourism context.

Considering that a positive attitude towards historical heritage sites could be increased by enhancing the appeal of heritage tourism products and their quality, along with those of related services, heritage managers, tourism promoters and destination managers should ensure this by carefully selecting what is communicated to their publics about historical heritage sites and communicating this to the public in the most effective ways. This could be achieved particularly by building effective marketing strategies based on the salient beliefs of domestic tourists, particularly in the heritage tourism context. Marketing strategies should be aimed at providing the relevant stimuli for attitude formation or favorable visit intention amongst visitors.

Significant individuals in the country's governmental, political, religious and social circles should be engaged (by heritage management, tourism promoters and destination managers) to actively participate in domestic heritage tourism and in the promoting of heritage attractions, e.g., cabinet ministers, chief executive officers, political leaders, religious, sports personalities and celebrities.

Roads and other accessibility facilities should be improved to facilitate domestic tourists' access to HHSs as destinations of choice.

1. Signposts should be erected in prominent positions to indicate the direction and locations of HHSs to enhance knowledge about their position and what they offer.
2. To enhance public knowledge about the geographical location of historical heritage sites and what they offer, high-quality and truthful information about them needs to be made readily available on all major digital platforms, including strategic influential websites and promotional networks.
3. Incentive holidays—the government of Kenya should offer an incentive to ensure people holiday within the country and in heritage destinations, rather than wildlife areas and beaches.
4. The government and other employers in the country should, as much as possible make weekends non-working days for their employees to have free time to tour the country. Alternatively, they should deliberately allow them off days, paid leave days, and paid holidays as incentives for domestic heritage tourism.
5. It is essential to stress the critical impact of perceived behavioral control on building tourists' patronage towards heritage destinations, because this factor presents a robust case for establishing interventions that create a sense of control and power for potential tourists and can also serve sustainability aims. As financial resources are one of the perceived action control factors for heritage tourism visits, increasing the salaries and wages paid to employees could go a long way in fostering domestic heritage tourism.

## 6. Conclusions

The present study successfully developed a comprehensive TPB model that accounts for the critical constructs in explaining domestic tourists' decisions to visit historical heritage attractions. Hence, the TPB was validated with respect to heritage visitation in a domestic tourism context. This study further extended the TPB model by adding motivation and perceived safety and security in the domestic heritage tourism context. It established that the model and its variables were applicable and more efficacious in directly predicting visit intention to historical heritage attractions. Adding motivation, and perceived safety and security improved the predictive power of the original TPB model.

These findings have generated substantial evidence related to all the hypotheses concerning domestic tourists' intentions to patronize historical heritage attractions. It also revealed that perceived behavioral control has the highest impact on domestic tourists' intentions to visit historic heritage attractions, whereas subjective norms had the least impact on the expanded TPB model. This underscores the need to emphasize these five factors more in marketing campaigns and other promotional strategies aimed at increasing domestic tourist visits to historical heritage attractions. This theoretical knowledge is important for academics, policy and practice with regard to heritage tourism, where all efforts and strategies should be aimed at increasing visit intention and sustainable visitor behavior to tourist attractions and destinations.

*Limitations and Future Research*

The current research widens our knowledge in relation to domestic tourists' tendency to visit historical heritage attractions with reference to the Kenya coast region. However, for this study, tourists' intention to visit these attractions was examined, instead of their actual visit behavior. Generally, it is believed that people are likely to overstate their intentions to engage in behavior [81], visiting heritage sites included. Therefore, it is hereby acknowledged that using a longitudinal approach would yield a more beneficial outcome for examining the actual visit behavior of domestic tourists who have indicated a willingness to visit the sites in future.

Decline in beta values of the three TPB variables in the second extended model suggested that the two variables added to the model could have had a moderating effect on attitude, subjective norm, and perceived behavioral control with respect to determining visit intention. This moderating effect needs to be investigated further.

A comparison of the level of behavioral intention to actual behavior revealed a significant difference, suggesting that not all intentions translate into actual behavior. Other factors, apart from visit intention, could be responsible for actual visits to historical heritage sites. There is a need to identify them.

**Author Contributions:** Conceptualization, P.O.O. and V.S.; methodology, P.O.O. and V.S.; software, P.O.O.; validation, P.O.O. and V.S.; formal analysis, P.O.O. and V.S.; investigation, P.O.O. and V.S.; resources, P.O.O.; data curation, P.O.O. and V.S.; writing—original draft preparation, review and editing, P.O.O. and V.S.; visualization, P.O.O. and V.S.; supervision, V.S. All authors have read and agreed to the published version of the manuscript.

**Funding:** This research received no external funding.

**Institutional Review Board Statement:** The study was conducted in accordance with the Declaration of Helsinki and approved by the Ethical Committee of the Hungarian University of Agriculture and Life Sciences Doctoral School of Economic and Regional Sciences (protocol code 12/2023 and 1/9/2023).

**Informed Consent Statement:** Informed consent was obtained from all subjects involved in the study.

**Data Availability Statement:** Data are contained within the article.

**Acknowledgments:** We acknowledge the financial support of the Institute of Agriculture and Food Economics and the Program for Excellent Researchers at the Hungarian University of Agricultural and Life Sciences (MATE).

**Conflicts of Interest:** The authors declare no conflicts of interest.

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
