# Peer review of "Behavioral Intention in Domestic Heritage Tourism—An Extension of the Theory of Planned Behavior"

_sustainability, doi:10.3390/su16020521_

Round 1

Reviewer 1 Report

Comments and Suggestions for Authors

The paper titled "Behavioral intention in domestic heritage tourism-an extension of the Theory of Planned Behavior" explores domestic heritage tourism in Kenya. It uses a broad base of background literature and quantitative methodology to investigate the phenomenon from the perspective of intentions to travel. The rationale given for the research is motivated by the need to use an extended model of TPB in this context, including motivation, perceived safety and security as predictors of visit intention. The manuscript is well written and structured. It focuses on domestic heritage tourism in Kenya for which there exists a gap in current research. The value in the study is in detecting possibilities for future work within the same context and goals, but with more long-term research approaches, and the need to identify factors for actual visits to heritage sites.

Author Response

Dear Reviewer, 

Thank you very much for your encouraging words. We highly appreciate it. 

Reviewer 2 Report

Comments and Suggestions for Authors

The topic is very interesting and important.

The structure of the paper is logical and in proper manner.

The hypotheses are checked and some conclusions and suggestions are done.

The methods used in the article enable to get the required results.

References 1-8 are not written in proper manner, please check them.

Author Response

Dear Reviewer, 

Thank you for your encouraging words. We appreciate it. 

The first 8 references (rows) were only the samples, and we deleted them. Thank you for the useful comment. 

Reviewer 3 Report

Comments and Suggestions for Authors

It appears to have been written without a great deal of familiarity with the requirements of the journal concerning the way to write the references. References should be reviewed. Reference in line 364, must be revised ( Hair et al. (2010). The authors should align the references to the  journal specifications. 

The references used are sufficient. However, the authors could cite more research in the last 5 years.

Information about the year about when the questionnaires were distributed should be presented ( from December to March) - add year.

No other revisions are needed. The topic is relevant and of high value.

Author Response

Dear Reviewer,
Thank you for the supporting words and the warning about the compilation of
references not as expected. Presumably, a small technical problem occurred during
copying and saving. We improved the references and added the years of the survey
to the manuscript as you can see in the modified version.

Reviewer 4 Report

Comments and Suggestions for Authors

The paper empirically tests the frame of Theory of Planned Behaviour (TPB) on the visit of historical heritage sites in Kenya.

The paper is well structured and uses extensive survey data, in order to test the model. In addition, the authors enriched the model with the inclusion of two new components: motivation and perceived safety and security as predictors of visit intention.

My opinion on the quality of the manuscript is very good, even if two issues have to be addressed before the final acceptance.

1.     My first concern relates to the representativeness of the sample. The distribution by gender is balanced, but I suspect that the distribution by education is not representative with more educated individuals that are overrepresented. This could be acceptable, because of the visit of historical heritage sites is more likely to be obtained from higher educated respondents. Anyway, more words should be spent on this point. Moreover, a description of the characteristics of the respondents that were dropped out from the sample has been done in order to capture sources of selection bias.

2.     My second concerns is on the performance of the second model (with the addicted variables, that is motivation and perceived security) with respect to the traditional one. The authors mention a small increase in the R2 (from 0.47 to 0.49) as stated at p. 16. This is not satisfactory at all for two reasons: the model with higher R2 is also less parsimonious, because of it integrates more variables and a high degree of complexity which causes a certain lost of the degrees of freedom. The attention should be posed to Adjusted R2 or similar measures of goodness of fit.

Author Response

Dear Reviewer, 

Thank you for your supporting opinion, on where you find the quality of our manuscript very good. 

Our point-by-point responses to your valuable comments are as follows: 

  1. We share your opinion that the visit to historical heritage sites is more likely to be obtained from higher educated respondents and we were aware of this characteristic even before starting the research. Precisely for this reason, we did not aim to collect data reflecting the representativeness of the Kenyan population, and a database showing the distribution of local heritage visiting tourists by education is not available. To reduce the selection bias, we continued random selection at each location and instead tried to select a representative location for the survey.
  2. We agree with the reviewer considering that the inclusion of more variables can most likely increase the rise of R2. That is why we did not formulate it as a hypothesis. The increase shows, however, that the included variables have a positive effect on the fit of the corrected model, and the results within the model support our hypothesis.

Reviewer 5 Report

Comments and Suggestions for Authors

It was with great interest that I read the article “Behavioral intention in domestic heritage tourism– an extension of the Theory of Planned Behavior”. This subject is actual and very pertinent, taking in account the increasing need to diversify the tourist offer in order to respond sustainably to the needs of current demand.

This article complies with almost all the established norms for the articles of this publication. The objectives are clearly presented. The research model is well built and clear, and the research hypotheses are interesting and well deduced from the literature review. The manuscript contains almost all the elements required in a scientific article. Bibliographic reference includes both older classic titles and current references. However, please note the followings:

1.There is not a theoretical background to support the study. Some literature review is done in the introduction but the relevance of the topic deserves its own literature review section separate from the Introduction. In my view, authors should make an effort to follow the traditional structure.

2. Materials and Methods section: the authors mention the months in which the study was carried out, but do not clarify when (year).

3. References: remove the first references that remains from the template. Review the references as some do not comply with the journal's rules.

Author Response

Dear Reviewer,

Thank you very much for your encouraging words, especially since you underline the importance of the topic from the aspects of sustainability.  Your suggestions were very helpful, and according to the order of them, we collected our answers together:

  1. We separated the introduction from the theoretical background as you suggested. Hopefully, this form not only highlights the relevance of the topic but can also serve to provide clarity.
  2. The years of the survey have been included  (2021/22).
  3. We deleted the first 8 sources (template) from the literature list and corrected other items according to the journal rules. More than half of the references are younger than 10 years, and 26,7% come from the last 5 years.  We updated 2 sources (nr. 57, 79) and added a new one (nr. 82).